

# Artificial reefs and marine protected areas: a study in willingness to pay to access Folkestone Marine Reserve, Barbados, West Indies

Anne E. Kirkbride-Smith[1], Philip M. Wheeler[2] and Magnus L. Johnson[1]

[1] School of Environmental Sciences, University of Hull, Kingston Upon Hull, United Kingdom
[2] Department of Environment, Earth and Ecosystems, The Open University, Milton Keynes, Buckinghamshire, United Kingdom

## ABSTRACT

Artificial reefs in marine protected areas provide additional habitat for biodiversity viewing, and therefore may offer an innovative management solution for managing for coral reef recovery and resilience. Marine park user fees can generate revenue to help manage and maintain natural and artificial reefs. Using a stated preference survey, this study investigates the present consumer surplus associated with visitor use of a marine protected area in Barbados. Two hypothetical markets were presented to differentiate between respondents use values of either: (a) natural reefs within the marine reserve or (b) artificial reef habitat for recreational enhancement. Information was also collected on visitors' perceptions of artificial reefs, reef material preferences and reef conservation awareness. From a sample of 250 visitors on snorkel trips, we estimate a mean willingness to pay of US$18.33 (median—US$15) for natural reef use and a mean value of US$17.58 (median—US$12.50) for artificial reef use. The number of marine species viewed, age of respondent, familiarity with the Folkestone Marine Reserve and level of environmental concern were statistically significant in influencing willingness to pay. Regression analyses indicate visitors are willing to pay a significant amount to view marine life, especially turtles. Our results suggest that user fees could provide a considerable source of income to aid reef conservation in Barbados. In addition, the substantial use value reported for artificial reefs indicates a reef substitution policy may be supported by visitors to the Folkestone Marine Reserve. We discuss our findings and highlight directions for future research that include the need to collect data to establish visitors' non-use values to fund reef management.

Corresponding author
Anne E. Kirkbride-Smith,
aesmith239@btinternet.com,
A.Kirkbride-Smith@hull.ac.uk

## INTRODUCTION

Coral reefs are of significant economic value to the scuba diving and snorkelling industries (*Brander, Van Beukering & Cesar*, *2007*) and via these water-based activities, reef tourism contributes millions of dollars annually to coastal regions (*Dixon, Scura & Van't Hof*, *1993*; *Cesar & van Beukering*, *2004*; *Sarkis et al.*, *2013*). A majority of reefs are located along the coastal strips of developing countries where people depend heavily on reef ecosystems for

their livelihoods (*Cesar, 2000*; *Cesar, Burke & Pet-Soede, 2003*; *Burke et al., 2011*). In St. Lucia and Tobago alone, direct spending by reef tourists in 2006 contributed an estimated US$91.6 and US$43.5 million to each economy, respectively (*Burke et al., 2008*). More recently, *Burke et al.* (*2011*) reported values for global reef tourism at US$50/ha/yr to US$1,000/ha/yr. In Bermuda, *Sarkis et al.* (*2013*) calculated the average total economic value of their coral reefs at US$722 million per year, from which US$406 million was related to coral reef tourism. Despite the value of coral reefs to coastal populations for marine recreation, shoreline protection and fisheries production, among others (*Moberg & Folke, 1999*), global reef decline continues as a result of various anthropogenic activities (*Halpern et al., 2008*; *Smith et al., 2016*).

Marine protected areas (MPAs) have become an effective means of conserving reef ecosystems from human impacts (*Halpern, 2003*; *Lester et al., 2009*), while still allowing for recreational use of resources including scuba diving and snorkelling (*Thurstan et al., 2012*). Considered by some to be the 'pinnacle' in marine conservation (*Thurstan et al., 2012*), an MPA is defined as "an area of sea especially dedicated to the protection and maintenance of biological diversity and of natural and associated cultural resources, and managed through legal or other effective means" (*Department of the Environment, 2013*). The last four decades have witnessed a proliferation of MPAs globally (*WDPA, 2013*). *Burke et al.* (*2011*) note that over two and a half thousand marine parks and equivalent protected areas have been designated to conserve coral reef habitats, amounting to 6% of the worlds coral reefs being managed. The many conservation benefits of MPAs are well documented (e.g., *Selig & Bruno, 2010*; *Johnson & Sandell, 2014*; *Leenhardt et al., 2015*), including an increase in the size and biomass of fish species (*Varkey, Ainsworthy & Pitcher, 2012*; *Caselle et al., 2015*; *Sciberras et al., 2015*). As a consequence, biological enhancement typically increases the attractiveness of marine parks to divers and snorkellers (*Barker, 2003*), though this in itself may cause a dilemma between protection and use of coral reef resources (*Thurstan et al., 2012*).

In general, MPAs manage visitor use of reefs through a system of zoning (*Day, 2002*; *Roman, Dearden & Rollins, 2007*) and by implementing carrying capacity measures (e.g., *Hawkins & Roberts, 1997*; *Brylske & Flumerfelt, 2004*; *Ríos-Jara et al., 2013*). Increasingly however, marine managers are investigating other ways of reducing the impacts of underwater recreational activities. Artificial reefs in MPAs have been envisaged as a potentially interesting management solution to deal with visitation levels to natural reefs (*Oh, Ditton & Stoll, 2008*), by providing additional habitat for marine biodiversity viewing (e.g., *Wilhelmsson et al., 1998*; *Van Treeck & Schuhmacher, 1999*; *Polak & Shashar, 2012*). This practice helps alleviate visitor pressures from sensitive or heavily used natural reefs (*Leeworthy, Maher & Stone, 2006*; *Polak & Shashar, 2012*; *Kirkbride-Smith, Wheeler & Johnson, 2013*) and may contribute significant revenues to local host economies (e.g., *Brock, 1994*; *Wilhelmsson et al., 1998*; *Dowling & Nichol, 2001*; *Johns et al., 2001*; *Johns, 2004*; *Pendleton, 2005*; *Oh, Ditton & Stoll, 2008*). However, the use of artificial reefs for amenity enhancement has not been without past criticism (*Oh, Ditton & Stoll, 2008*). Such condemnation has largely been due to the ubiquitous use of 'materials of opportunity' for reef creation (*Stone et al., 1991*; *Tallman, 2006*), including car tyres (*Collins, Jensen & Albert, 1995*; *Collins et al., 2002*).

Nevertheless, well conceived artificial reefs may facilitate various management strategies within protected waters including influencing the location of recreational use (*Leeworthy, Maher & Stone*, *2006*; *Polak & Shashar*, *2012*) and visitor behavior, via scientifically-based interpretation materials (*Rangel et al.*, *2014*).

Despite the potential efficacies of MPAs (*Halpern & Warner*, *2002*; *Halpern*, *2003*; *Lester et al.*, *2009*), many fail to meet management objectives (*Burke, Selig & Spalding*, *2002*; *Burke & Maidens*, *2004*; *Wells*, *2006*; *Burke et al.*, *2011*; *De Santo*, *2013*), are severely under funded (e.g., *Alder*, *1996*; *Depondt & Green*, *2006*) and exist as 'paper parks' only (*Brandon, Redford & Sanderson*, *1998*; *Bruner et al.*, *2001*; *Bonham, Sacayon & Tzi*, *2008*; *Mora & Sale*, *2011*). Various funding mechanisms exist for MPAs including personal donations, lottery revenues, international assistance and government taxes (*Spergel & Moye*, *2004*). However, none of these mechanisms are wholly reliable. For instance, government taxes can be re-directed to responsibilities elsewhere (*Lindberg*, *2001*), especially in times of economic difficulties (*Spergel & Moye*, *2004*). Reef-based tourism is considered to be a lucrative means of financing protection of marine parks (e.g., *Dharmaratne, Sang & Walling*, *2000*; *Depondt & Green*, *2006*; *Peters & Hawkins*, *2009*), through the recovery of user fees from visitors. Techniques, including the contingent valuation method of willingness to pay (WTP), can be used to determine the level visitors would contribute.[1,2] Fees collected, can increase the management capacity of parks through for example; education, scientific monitoring and enforcement (*Hime*, *2008*; *Uyarra, Gill & Côté*, *2010*), collectively helping sustain future conservation of reefs. However, many marine reserves remain free to use, or charge a nominal entrance fee (*Peters & Hawkins*, *2009*; *Terk & Knowlton*, *2010*), despite evidence that in some circumstances, user fees could increase substantially with little impact on visitor numbers (*Thur*, *2010*).

*Bryant et al.* (*1998*) and *Burke et al.* (*2011*) emphasize the need for countries harbouring coral reefs to conduct applied valuation techniques to help underpin decision and policy-making. An integral part of WTP studies is to discern what motivates people to donate funds. The non-economic motives behind WTP for biodiversity conservation have been explored (*Martín-López, Montes & Benayas*, *2007*) with results proposing familiarity and biophilia as having a marked effect on payment attitudes. Some authors (e.g., *Cooper, Poe & Bateman*, *2004*; *Spash*, *2006*) suggest that intrinsic value is the main motivator explaining visitor's choice to contribute, as is bequest value that benefits future generations (*Hargreaves-Allen*, *2010*). Researchers have also sought to establish what factors influence how much visitors are prepared to pay. Studies indicate that users of reefs (usually divers and snorkellers surveyed) are willing to allocate more money for an increase in the abundance or quality of a specific reef attribute or group of attributes (e.g., *Rudd & Tupper*, *2002*; *Schuhmann, Casey & Oxenford*, *2008*; *Polak & Shashar*, *2013*). Additionally, the opportunity of viewing charismatic mega-fauna, including marine turtles and whale sharks, is greatly valued (*Hargreaves-Allen*, *2010*; *Schuhmann et al.*, *2013*; *Farr, Stoeckl & Beg*, *2014*). Conversely, studies have noted losses in consumer surplus relating to the demise of coral reefs.[3] For example, *Doshi et al.* (*2012*) reported a reduction in divers' welfare identified by their decrease in WTP for bleached coral reefs.

[1] Contingent valuation is a survey-based methodology for eliciting values people place on goods, services and amenities (*Boyle*, *2003*).

[2] WTP is defined as, "the maximum amount a person is willing to pay for a good or service" (*Waite et al.*, *2014*).

[3] Consumer surplus is the difference between the price that consumers pay and the price that consumers are willing to pay (*Bateman et al.*, *2002*).

**Table 1  Selected papers and key findings of WTP studies to access coral reefs in MPAs.**

| Author(s) (year) | Location | Users surveyed | Per | Value per user[a] WTP mean | Median | Suggested fee |
| --- | --- | --- | --- | --- | --- | --- |
| *Dixon, Scura & Van't Hof (1993)* | Bonaire | Divers only | Annum | $27.40 | $20 | $10 |
| *Spash (2000)* | Jamaica | Locals & tourists | Annum | $25.89 | $2.87 | N/R |
| *Spash (2000)* | Curaçao | Locals & tourists | Annum | $25.21 | N/R | N/R |
| *Arin & Kramer (2002)* | Anilao, Philippines | Divers & snorkellers | Visit | $3.70 | $3 | $4 |
| *Arin & Kramer (2002)* | Mactan, Philippines | Divers & snorkellers | Visit | $5.50 | $5 | $5.50 |
| *Arin & Kramer (2002)* | Alona, Philippines | Divers & snorkellers | Visit | $3.40 | $3 | $4 |
| *Mathieu, Langford & Kenyon (2003)* | Seychelles | Divers & snorkellers | Visit | $12.20 | N/R | $12.20 |
| *Seenprachawong (2003)* | Phi Phi, Thailand | Divers & snorkellers | Visit | $7.18 | N/R | $1 |

Notes.

[a]Reported in year of study in US dollars.

N/R,  not recorded in original paper.

Numerous researchers (e.g., *Dixon, Scura & Van't Hof, 2000*; *Arin & Kramer, 2002*; *Barker, 2003*; *Mathieu, Langford & Kenyon, 2003*) have undertaken contingent valuation surveys to measure visitors' WTP for marine park entry (Table 1).  In a meta-analysis detailing 18 studies, *Peters & Hawkins* (*2009*) found an overwhelming approval of users to pay marine park access fees, or an increase in fees, where charges currently existed. Additionally, there is evidence that user fees can generate sufficient funds to cover a significant share of MPA operating costs (*Spergel & Moye, 2004*). For example, in Australia's Great Barrier Reef Marine Park, tourist-based user fees of US$5 million contributed around 20% of the budget of the park authority in 2002/2003 (*Skeat & Skeat, 2003*). On Bonaire, user fee collections of around US$1 million represented 93% of the income required to operate the National Marine Park in 2008 (*STINAPA, 2009*; *Uyarra, Gill & Côté, 2010*).

To date, there has been a clear emphasis on measuring the consumer surplus of visitors' recreational use of natural reefs (*reviewed in Peters & Hawkins, 2009*). In contrast, only a handful of contingent valuation studies appear to have measured visitors' consumer surplus relating to recreation-orientated artificial reefs (*Bell, Bonn & Leeworthy, 1998*; *Ditton & Baker, 1999*; *Johns et al., 2001*; *Johns, 2004*; *Crabbe & McClanahan, 2006*; *Oh, Ditton & Stoll, 2008*; *Hannak et al., 2011*; *Chen et al., 2013*). However, none of these studies used marine park user fees as the payment vehicle to estimate consumer surplus, and just three papers (*Johns et al., 2001*; *Johns, 2004*; *Oh, Ditton & Stoll, 2008*) estimated recreational values of artificial and natural reefs in the same locality. To address this dearth of information, a valuation study was developed that encompassed both artificial and natural reef habitats within a MPA.

The main purpose of this analysis was to investigate the present consumer surplus associated with visitor use of a MPA in Barbados, using the contingent valuation method. We discuss our findings with relevance to visitors funding reef conservation and highlight the potential that reserves and artificial reefs have for symbiotic partnerships in coral reef management.

## METHODS

### Study setting

This study was conducted on the west (leeward) coast of Barbados (13°10′N, 59° 32′W) between the months of July to August 2013, over an 18 day period. Akin to many Caribbean islands, the tourism appeal of Barbados depends on its coastal environment. Coral reefs fringing the south-west coast (*Lewis*, *1960*) provide a diversity of recreational opportunities including scuba diving, snorkelling and sub-marine viewing. *Schuhmann, Casey & Oxenford* (*2008*) estimate that between 30,000 and 50,000 divers visit the island per year and the *Inter-American Biodiversity Information Network* (*2010*) report a further 176,600 visitors participating in snorkel trips. As a way of diversifying the marine tourism industry, several artificial reefs have been deployed along the south-west coast (*Agace*, *2005*).

One small MPA (2.1 km$^2$) the Folkestone Marine Reserve, is located in the parish of St. James on the western side of the island (*Cumberbatch*, *2001*). The reserve extends for 2.2 km along the coastal fringe and stretches outwards between 660 and 950 m offshore (Fig. 1). Legislated in 1981 (*Cumberbatch*, *2001*), the marine reserve protects 0.32 km$^2$ of accessible fringing, patch and bank reef (*Inter-American Biodiversity Information Network*, *2010*) and nesting sites of the endangered hawksbill turtle *Eretmochelys imbricata* (*Horrocks & Scott*, *1991*; *Beggs, Horrocks & Krueger*, *2007*). A small artificial reef consisting of a disused barge (approximately 8 m long), that provides a site for instructor-led dives and for snorkellers, is situated within the reserve (Fig. 1). Encompassing just 11% of the coastline (*Cumberbatch*, *2001*), the reserve attracts multiple stakeholders and represents the most heavily used recreational space in Barbados (*Blackman & Goodridge*, *2009*), including approximately 7,000 scuba divers using the Folkestone reefs per year (*Inter-American Biodiversity Information Network*, *2010*). In anticipation of potential user conflict, the reserve has been divided into four distinct zones (*Cumberbatch*, *2001*) (Fig. 1). The sites used for this study were located within the Folkestone Marine Reserves Zone D—southern water sports zone (principally Sandy Lane patch reef and the disused barge—Site 1) and a site to the outside of the northern reserve boundary (Site 2), adjacent to the Lone Star reef (Fig. 1).

### Valuation method and related issues

In order to estimate maximum WTP, the surveys (Supplemental Information) adopted a payment card contingent valuation method. Other common response formats used to measure demands for non-market goods, are single- and double-bounded dichotomous choice and open-ended questioning techniques. All four valuation approaches are subject to some degree of bias (*Bateman et al.*, *2002*; *Boyle*, *2003*), though this can be reduced with the careful design and pre-testing of surveys (e.g., *Boyle et al.*, *1998*). Despite various biases, each of these stated preference techniques uses hypothetical market scenarios to discern a respondent's likely behaviour under various conditions of either WTP, or willingness to accept, for an increase/decrease in a public good. In the case of the payment card approach, it uses an ordered set of threshold values that respondents are asked to peruse and indicate the highest amount they are willing to pay. *Bateman et al.* (*2002*) and *Boyle* (*2003*) outline the various advantages of payment cards including the avoidance of anchoring and 'yea saying' to a sole bid presented (a problem in dichotomous choice) and the avoidance of
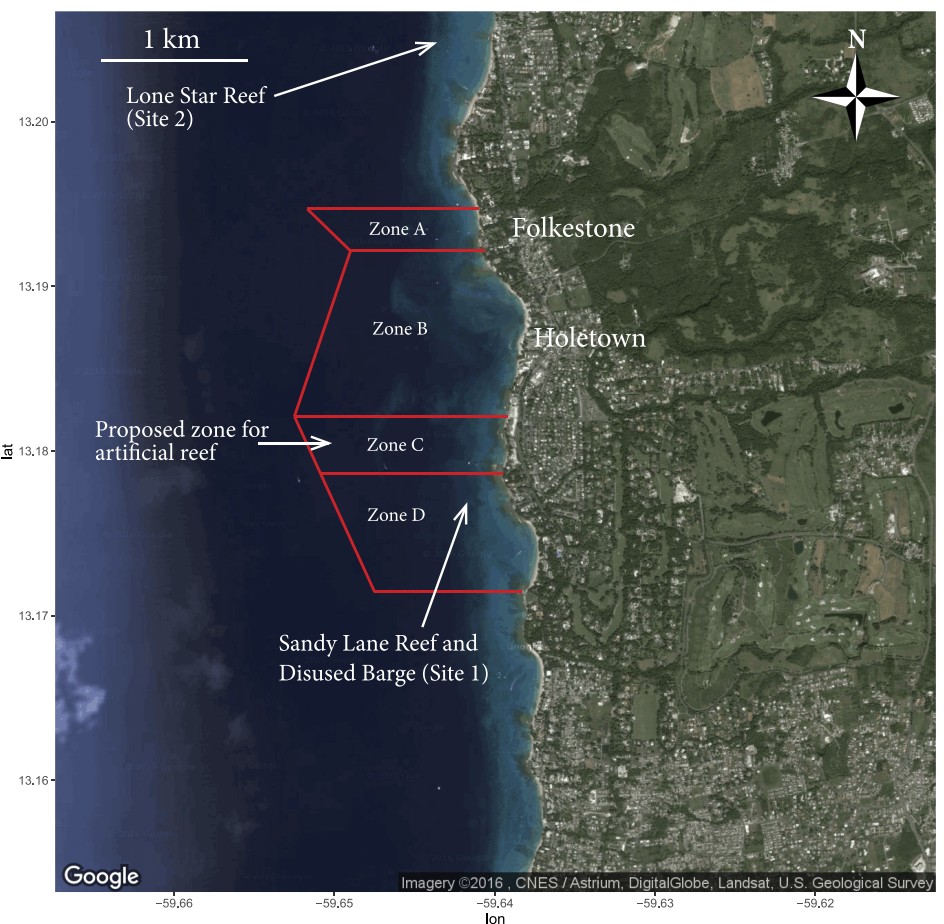

**Figure 1** **The Folkestone Marine Reserve, Barbados.** Map outlining boundary of marine protected waters and locations of study sites and proposed artificial reef. Zone A: Scientific Zone (196 m$^2$); Zone B: Northern Water Sports Zone (819 m$^2$); Zone C: Recreational Zone (460 m$^2$); Zone D: Southern Water Sports Zone (625 m$^2$) (*Modified from Google Earth. Map data: Google, CNES/Astrium, DigitalGlobe, Landsat, U.S. Geological Survey*).

starting point bias. In addition, *Mitchell & Carson* (*1989*) suggest payment cards can assist in reducing non-response rates and eliminate the need for prompting by the interviewer. They have also been shown to yield WTP estimates that are more conservative than those generated using other stated preference techniques (*Champ & Bishop*, *2006*; *Thur*, *2010*). Payment cards are however, subject to specific forms of bias relating to the design configuration in range of monetary values and size of intervals chosen (*Bateman et al.*, *2002*). In fact, in payment card data, the true WTP value is thought to lie between the bid amount chosen and the next highest value up on the payment card (*Cameron & Huppert*, *1989*; *Bateman et al.*, *2002*; *Boyle*, *2003*). Thus intervals rather than 'point' valuations are used in most statistical models.

## Survey design and data collection

An initial site visit to the Folkestone Marine Reserve was conducted in 2012, to determine if any entrance fee payment was already in place (of which there were none) and to

determine visitor trips/user patterns within the reserve. Additionally, an informal focus group consisting of snorkellers and divers was held to ascertain the range of bid values to be used in the data collection instrument. Two versions of the survey were produced; one aimed at valuing artificial reefs and the second aimed at valuing natural reefs. Both instruments were identical with the exceptions of sentence three and the word 'artificial reef' in sentence five of the artificial reef valuation question (presented below), which were omitted from the natural reef script. The payment vehicle used was a daily, per person user fee into the marine reserve. The final survey consisted of 46 questions divided into five sections. A majority of the questions were closed-ended, as *Champ* (*2003*) suggests this format helps avoid respondent fatigue and simplifies statistical analysis in WTP studies.

The first section explored respondents demographic characteristics that included number of years spent in education, country of residence and age. In this section also, participants were asked questions relating to their length of stay in Barbados and about any previous visits to the island. In the second section, visitors were questioned about their marine recreation participation. A 5-point Likert rating scale (range: very experienced to very poor) was presented to establish their snorkelling proficiency. To gauge the experience of those participants that had scuba diving ability, we asked for the number of dives they had logged in their diving history. A similar scale (range: very satisfied to very dissatisfied) was also used to assess visitor satisfaction with snorkelling, and if applicable, diving on the island. The final question in section two assessed which marine related activities respondents had undertaken during their present stay. In the third part of the survey, the hypothetical valuation scenario was presented to establish each visitor's WTP bid amount. The valuation script (Supplemental Information) contained background information pertinent to the reefs within the reserve and the challenges encountered in managing them. A laminated map of the reserve (Fig. 1) was shown to each visitor prior to the WTP question being asked, as were photos of common species found within the reserve. Additionally, in the artificial reef survey, laminated cards of popular artificial reef materials were presented (Supplemental Information). The exact wording of the valuation question presented in the artificial reef survey was:

*Today, no Folkestone Marine Reserve fee is paid by you to visit the coral reefs and marine species within this protected area. All funding to conserve the reefs here is sourced elsewhere. There is a proposal to develop one or more artificial reefs within the marine reserve for both snorkelling and diving (show map and explain). A visitor user fee (held in a trust fund) would be used to help manage and maintain the artificial reefs within the reserve. With this in mind, I am going to show you a set of numbers in US dollars. Please consider your total trip costs for this visit and tell me; what is the maximum you would be willing to pay '**over and above your present trip costs**' as a daily user fee to recreate in the Folkestone Marine Reserve?*

The survey presented 12 payment values in ascending order (*Champ*, *2003*) from US$0 to US$60 (Table 2), from which respondents were asked to choose a value (or to specify another amount if above US$60) as an indication of their WTP to help manage and maintain the reefs within the reserve. Section three of the survey also included follow-up questions exploring the rationale given for a bid value, or if a zero bid was given, the reason for that particular choice. We also asked respondents which type of organization they
**Table 2** Interval selection frequencies of WTP bids (daily, per person).

| Interval (US$) | All data ($n = 250$) | Raw frequency (%) AR data ($n = 125$) | NR data ($n = 125$) |
| --- | --- | --- | --- |
| 0 | 7 (2.8) | 4 (3.2) | 3 (2.4) |
| 2–5 | 4 (1.6) | 3 (2.4) | 1 (0.8) |
| 5–8 | 22 (8.8) | 12 (9.6) | 10 (8.0) |
| 8–10 | 26 (10.4) | 11 (8.8) | 15 (12.0) |
| 10–15 | 70 (28.0) | 35 (28.0) | 35 (28.0) |
| 15–20 | 43 (17.2) | 16 (12.8) | 27 (21.6) |
| 20–25 | 42 (16.8) | 26 (20.8) | 16 (12.8) |
| 25–30 | 12 (4.8) | 7 (5.6) | 5 (4.0) |
| 30–40 | 11 (4.4) | 8 (6.4) | 3 (2.4) |
| 40–50 | 6 (2.4) | 1 (0.8) | 5 (4.0) |
| 50–60 | 4 (1.6) | 1 (0.8) | 3 (2.4) |
| >60 | 3 (1.2) | 1 (0.8) | 2 (1.6) |

**Notes.**

AR, Artificial reef; NR, Natural reef.
Figures in parenthesis are percentages.

would prefer to manage the user fee revenues and enquired about any concerns relating to the management of funds raised. The fourth section of the survey was used to query respondents on their knowledge and use of artificial reefs, both in Barbados and elsewhere in the world. We included a specific question to identify respondents preferences, placed in rank order, relating to types of materials used for artificial reef creation. At this point of enquiry, three laminated cards with images of artificial reefs were shown to individuals (Supplemental Information). Three questions were also embedded in section four to help capture each visitor's environmental awareness and general concern for reefs and the marine environment. The final part of the survey aimed to establish respondent's prior and current experience(s) of the Folkestone Marine Reserve. We asked visitors to use a 5-point Likert rating scale (range: very good to very poor) to rate the quality of the seawater, coral and fish life encountered on their present trip. A question was also used to establish what marine life visitors had viewed whilst underwater. Finally, respondents were requested to score their overall experience of the reserve on a 4-point Likert rating scale (range: exceeded expectations to not satisfied expectations) after which visitors were asked to clarify if they had plans to return to the reserve in future.

A preliminary test of the survey ($n = 20$) was conducted in Barbados on the target population and changes made accordingly, prior the main data collection period. _Dharmaratne & Brathwaite_ (_1998_) emphasize the importance of choosing respondents familiar with the good being valued, thus the sample frame population consisted of snorkellers and/or divers with prior experience of either activity. In addition, English speaking overseas tourists of any nationality, between the ages of 18 to 70 years visiting the reserve, were a requirement. As very few Barbadian residents snorkel or scuba dive (_Inter-American Biodiversity Information Network_, _2010_), they were not included in the surveying process.

Visitors to the Folkestone Marine Reserve were approached on board Tiami catamaran cruise trips (www.tiamicruises.com). These 5 h snorkel trips, at a cost of US$85 per person, provide visitors with two 30 min snorkel stops (Fig. 1) and a beach visit. A sampling technique was chosen to examine the population by approaching every other seated tourist, moving systematically from the front to the rear of the catamaran. In view of the fact that interview context has been reported as a significant determinant of WTP (*Arrow et al.*, *1993*; *Hime*, *2008*; *Hargreaves-Allen*, *2010*) all interviews were conducted personally using the same location (i.e., on-board a Tiami catamaran) and after experiencing the reserves underwater environment. Each interview took approximately 20 min to complete. For consistency, the same two interviewers administered both surveys on a rotational (daily) basis, initially giving each respondent a short introduction to explain the reasons for the survey. Only one survey type was administered to each respondent. Prior to the bid valuation question being presented, it was emphasized that no user fee is currently imposed on visitors to the reserve. All visitors who participated in the survey gave their permission to use the results on an anonymous basis.

## Data analysis and WTP estimations

Responses were analyzed using SPSS (Version 19) and R (*R Development Core Team*, *2008*). To investigate differences between the responses given in survey 1 (artificial reef scenario) and survey 2 (natural reef scenario), we applied Chi-square tests with Yate's Continuity Corrections for categorical data, and Mann–Whitney $U$ tests (two-tailed) for continuous data. Variations in WTP were investigated for several variables (e.g., between snorkellers and divers and for Likert scale questions) using Mann–Whitney $U$ tests (two-tailed) and Kruskal–Wallis tests, where applicable. Consistent with the method adopted in *Fitzsimmons* (*2009*), a distinction was made between the experience level of participant divers, denoted by two categories; novice divers (<100 logged dives) and experienced divers (≥100 logged dives).

Data were screened for zero bids (US$0), with each bid individually assessed via the follow up questions, as to why the respondent was not prepared to pay. Mean and median WTP, prior to and after zero bid removal, were compared. Following *Bateman et al.* (*2002*), zero bids were excluded from the data prior to calculating mean and median WTP for all models. Significant differences between the two study populations were tested to ensure that specific characteristics of the sample (e.g., age and gender) had not been systematically biased. Standard errors and 95% confidence intervals of estimates of WTP were calculated using bootstrapping (*Kling & Sexton*, *1990*) based on 1,000 replications.

## Econometric analysis

The theoretical foundation of WTP is based on the assumption that individuals derive utility from consumption of an environmental public good and are assumed to maximize their utility given income and commodity prices. WTP is hypothesized to be influenced by a number of independent variables (*Arin & Kramer*, *2002*) represented by the vector $\boldsymbol{x}$.

$$\mathrm{WTP}_i = \beta' x_i + \varepsilon_i$$

where $\beta$ is a vector of slope parameters to be estimated and $x_i$ is a vector of observations on the explanatory variables for individual $i$. The error term $\varepsilon_i$ is assumed to be normally distributed.

Payment card data were analyzed using interval regression (*Bateman et al.*, *2002*), as it is thought that the true payment value given lies between the value chosen and the value bounding the upper interval of that category (*Cameron & Huppert*, *1989*). Thus for the payment card sample, a maximum likelihood estimation (MLE) procedure was used (*Cameron & Huppert*, *1989*) that accommodates the intervals, that is the probability that WTP falls in the range defined by the lower limit $t_{li}$ and the upper limit $t_{ui}$, represented by the adjacent payment card value given by;

$$\Pr(\log w_i \subseteq (\log t_{li}, \log t_{ui})) = \Pr(\log t_{li} - X_i'\beta)/\sigma < z_i < \Pr(\log t_{ui} - X_i'\beta/\sigma),$$

where $z_i$ is the standard normal random variable. *Arin & Kramer* (*2002*) note that because the probability given by the latter equation can be written as the difference between two standard cumulative densities a likelihood function can be defined over the parameters $\beta$ and $\sigma$. In the study, interval boundary parameters were estimated using the survival package (*Therneau*, *2014*).

For comparison, an ordinary least squares regression model was also applied. In the latter model, the precise mid-point of each interval category is used as the dependent variable of WTP. Normality is assumed for the regression models (*Cameron & Huppert*, *1989*), with a *lognormal* conditional distribution proposed as a first approximation. Many researchers have adopted (*Cameron & Huppert, 1989*) method in WTP studies using payment cards (e.g., *Arin & Kramer*, *2002*; *Blaine et al.*, *2005*; *Mahieu, Riera & Giergiczny*, *2012*; *Yang, Hu & Liu*, *2012*), as one of the advantages is that value estimates can be interpreted in a straightforward manner (as apposed to log transformed data). Also, by using both interval regression and an ordinary least squares model, it helps validate the payment card range presented and serves as an ad hoc check of the normality assumption. The stepwise backward elimination method was employed for both regression models to investigate the effects of 12 independent predictor variables (Table 3) on visitors' total WTP. Variables that did not yield covariates significant at $\leq 10\%$ level were excluded from the final model.

## RESULTS

### Visitor and holiday characteristics

Two hundred and fifty surveys were completed during the study period divided equally between the two reef scenarios ($n = 125$ for each survey). An almost equal sex ratio (51% female) was recorded from both surveys combined. The majority of visitors resided in the United Kingdom (72%), followed by the United States (12%), with five additional countries (Canada, Brazil, Norway, Italy and the Caribbean Island States) making up the sample. The mean and median age of respondents was 38 ($\pm 13.6$ s.d.) and 40 years respectively, with an age range of 18–69 years recorded. The total number of years visitors had spent in education ranged from 11 to 27 years with the average length being 16 ($\pm 3.3$ s.d.) years. Over a third (38%) of those surveyed, were repeat visitors to Barbados with a

**Table 3 Descriptions of the explanatory variables.**

| Variable | Description |
| --- | --- |
| Age | Continuous: the age of the respondent |
| Gender | Discrete: 1 = male, 0 = female |
| Education | Continuous: number of years the respondent has spent in education |
| Barbados_visits | Continuous: number of visits to Barbados |
| Env_concern | Continuous: level of environmental concern: 1 being the least concerned, 10 being the most concerned |
| Catamaran_cruise | Continuous: how many catamaran cruises undertaken in the Folkestone Marine Reserve? |
| Dived_FMR | Discrete: if the respondent had dived in the Folkestone Marine Reserve, 1 = yes, 0 = no |
| Species_view | Continuous: number of species mentioned in response to open ended question to the no. of species encountered |
| Satisfaction_trip | Discrete: did the snorkel trip satisfy expectations? 1 = yes, 0 = no |
| Fish_life | Discrete: if the respondent rated the fish life viewed as good, 1 = yes, 0 = no |
| Coral_life | Discrete: if the respondent rated the coral life viewed as good, 1 = yes, 0 = no |
| Seawater_quality | Discrete: if the respondent rated the seawater quality as good, 1 = yes, 0 = no |

mean of 3 ($\pm$3.9 s.d.) visits (including the present one). The number of nights being spent on the island ranged from 2 to 30 nights, with the majority (50%) of respondents having an average duration of 12 ($\pm$3.9 s.d.) stop-overs. Group differences investigated between survey 1 and survey 2 identified one variable; *Age* being statistically different between the two surveys ($U = 6,173$, $z = -2.206$, $p \leq 0.027$, $r = 0.14$). Artificial reef survey participants were slightly older than natural reefs survey participants; means: 39 ($\pm$14.25 s.d.) and 36 ($\pm$12.7 s.d.) years, medians: 43 and 36 years, respectively. Data from the *Barbados Hotel & Tourism Association* (*2016*) for visitors to Barbados in 2013 were used to assess for sample representativeness. From the limited data available, tourist stop-over arrivals for that year suggest that our sample was over-represented by UK respondents. Additionally, no cruise ship tourists were available for interview.

## Marine recreation participation

Prior to the survey being administered, visitors had carried out 3.75 ($\pm$0.9 s.d.) activities whilst on vacation. The majority had relaxed on the beach (85%), swam (81%), snorkelled from the shore (39%), kayaked (21%) and scuba dived (12%). The majority of snorkellers described themselves as being average (50%) to very good (31%) at the sport, while 17% suggested they were poor and a further 2% very poor at snorkelling. Respondents that had scuba diving ability ($n = 76$), had an average of 32 ($\pm$86.81 s.d.) previously logged dives and a median of 10 dives [interquartile range: 2–25]. Seventy-four percent of the sample had been given a snorkelling and/or diving briefing at some point in their life. When visitors were asked to rate their satisfaction with snorkelling on the island in general, 83%

**Table 4** Respondents' WTP to access the Folkestone Marine Reserve (daily, per person) in US$.

| WTP scenario | N | Lower[a] bound CI | Mean ± 1SD | Upper bound CI | Median |
|---|---|---|---|---|---|
| All data (zero bids in) | 250 | 15.92 | 17.45 ± 11.30 | 18.96 | 12.50 |
| All data (zero bids out) | 243 | 16.62 | 17.96 ± 11.05 | 19.27 | 12.50 |
| Artificial reef data | 121 | 15.81 | 17.58 ± 9.96 | 19.52 | 12.50 |
| Natural reef data | 122 | 16.25 | 18.33 ± 12.06 | 20.73 | 15.00 |

Notes.
[a]Based on 1,000 replications.

was either satisfied (41%) or very satisfied (42%) with the experience, with the remainder being ambivalent. Respondents who had dived ($n = 39$) whilst visiting Barbados, were all either satisfied (66%) or very satisfied (34%) with their prior experiences.

## The Folkestone Marine Reserve WTP

A total of 7 zero bids (Table 4) for WTP were identified. Follow-up questions were asked to establish the reason why a zero bid was given. Four individuals were uncertain the money would be spent on reef conservation *per se*, while the remaining respondents were unsure their contributions would make any difference to the condition of the reefs in the Folkestone Marine Reserve.

Zero bids were removed and mean and median values calculated for pooled data and for each survey type (Table 4). Mean values were higher than median values for all estimates calculated. This was due to positive right skews in the WTP distributions. The removal of the few zero bids had a meager US$0.51 impact on mean WTP (Table 4), which did not bias the results. For pooled data, mean WTP (person/day) was estimated at US$17.96 with a lower bound of US$16.62 and an upper bound of US$19.27, at a 95% confidence interval. Visitors who participated in the natural reef survey, had a higher mean WTP of US$18.33, in comparison to mean values estimated for visitors presented with the artificial reef survey; US$17.58. The median value was also higher for the natural reef scenario (US$15) than for the artificial reef scenario (US$12.50). Differences in WTP between the two survey types were not significant ($U = 7,291$, $z = -.167$, $p \geq 0.867$, $r = .01$).

Table 5 shows differences in mean WTP for selected variables. Females had a significantly higher WTP of US$19.54, compared with a value of US$16.31 estimated for males. Visitors, who had viewed a turtle while snorkeling, had a value of US$19.59 compared with US$11.56 for those who had not viewed a turtle. This latter difference of US$8.03 was highly significant. Divers, who had experienced the underwater environment within the reserve prior to being interviewed, had a lower WTP of US$12.50, compared with divers visiting the reserve for the first time of US$18.55. Finally, repeat catamaran trip visitors had a significantly lower bid value of US$13.37, compared with individuals who were first time visitors to the reserve of US$18.45. From a point of interest, snorkellers and those who had diving ability had a very similar mean bid value of US$17.89 and US$16.45, respectively.

Motivations of respondents' WTP were explored. Most visitors (75%) reported that they would pay a user fee to help preserve coral reefs for future generations, followed by

**Table 5** Differences in respondents' mean WTP (US$) for selected variables.

| Variable | Structure | N | WTP | ±1SD | P value |
|---|---|---|---|---|---|
| *Gender* | Female/Male | 124/119 | 19.54/16.31 | ±11.89/9.89 | $p \leq 0.007$ |
| *Turtle* | Yes/No | 196/47 | 19.59/11.56 | ±11.50/5.52 | $p \leq 0.001$ |
| *Dived_FMR* | Yes/No | 24/52 | 12.50/18.55 | ±5.95/11.32 | $p \leq 0.002$ |
| *Repeat_trip* | Yes/No | 49/194 | 13.37/18.45 | ±8.12/11.74 | $p \leq 0.003$ |
| *Activity* | Snorkel/Dive | 167/76 | 17.89/16.45 | ±11.24/11.43 | $p \geq 0.842$ |

10% indicating it gave them genuine pleasure to contribute towards reef conservation. A motivator of being a 'moral duty' to contribute was also important among 8% of visitors. Of those who were willing to pay, 70% reported concerns over the legitimate use of monies collected for reef conservation, while the remaining 30% of visitors reported no concerns. Content analyses of the follow-up questions to understand these concerns revealed that most individuals were anxious that the funds raised would be spent elsewhere; typically on other government projects in Barbados. Respondents were also asked which type of organization they would prefer to manage the user fee revenues. An environmental non-governmental organization was clearly the most popular choice yielding 75% support, followed by the government of Barbados (13%) and public sector (3%), while 9% chose a mix of all three authorities. The question that queried respondents in relation to where they would prefer to see park fee revenues spent, yielded a high level of support for marine education/children's outreach programmes (47%) and for recreational artificial reefs (27%). Scientific monitoring also appeared important with 18% of respondents choosing this item. In contrast, land-based tourist facilities (1%) and marine reserve patrols (2%) seemed unimportant investments.

## Perceptions and use of artificial reefs and environmental concern

Artificial reef awareness was high amongst the population sampled with 69% having heard of the term artificial reef, and 82 respondents (34%) having either snorkelled or dived on an artificial reef previously. When asked to rate their experience of this type of reef, 79% of snorkellers and 88% of divers rated their prior experiences as good to very good. Additionally, 35 respondents had used local artificial reefs, the majority ($n = 29$) situated in Carlisle Bay and the remaining 6 individuals using the *SS Stavronikita*, the largest wreck to dive on in the Caribbean (*Agace*, *2005*).

Three reef material types were presented using visual aids (Supplemental Information). The most preferred material choice was a shipwreck (73%), followed by Reef Balls™ (as a snorkel trail) (17%), with underwater art chosen by only 10% (Fig. 2). Asked whether the creation of an artificial reef in the Folkestone Marine Reserve would encourage a repeat visit, 77% answered yes, 12% no and 11% were unsure.

When questioning respondents if they were a member of an environmental group, only 10% responded positively. In contrast, 83% of visitors read or watched on television topics about marine life and marine conservation. Respondents rated their level of concern relating to coral reefs and the marine environment (on a scale of 1–10, with 1 being the least concerned) with a mean and median value of 7 (±1.77 s.d.).

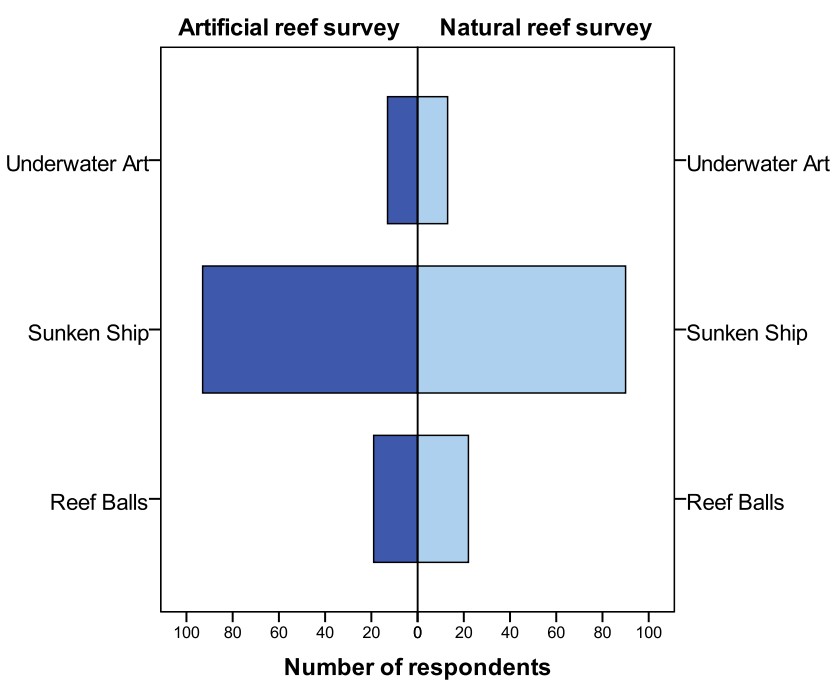

**Figure 2** **Respondents' preferences for type of artificial reef material for future use in the Folkestone Marine Reserve.** Sample size: $n = 250$.

## Experience of the Folkestone Marine Reserve

A fifth ($n = 49$) of respondents had previously visited the reserve on catamaran snorkelling cruises, with 1.84 ($\pm 2.63$ s.d.) former trips recorded. All respondents said they had snorkelled during these trips. Additionally, 24 respondents that had previously dived in the reserve, had conducted 4.88 ($\pm 4.31$ s.d.) dives there.

Respondents were asked to recall the number of 'species' viewed. The marine life noted in the study was; fish, coral, turtles, eels, manta rays and sea urchins. A majority of visitors recalled three species (3.4 ($\pm 1.11$ s.d.), median and mode = 3) with a maximum of six species seen, with no person being noted as viewing no marine life. The most common species recalled were fish, spotted by 95% of people, followed by a turtle noted by 80% of visitors.

Thirty-two percent of respondents had their expectations of the visit to the reefs exceeded and a further 55% were noted as being satisfied. Only 19 individuals said the trip had made no difference to them, while 8 visitors had not had their expectations satisfied. A significant relationship occurred between visitors' WTP and their level of satisfaction with the marine park (Kruskal-Wallis test; $x^2$ (3) = 12.32, $p \leq 0.006$). Further post hoc analysis revealed the two groups most dissatisfied/ambivalent with the trip (when combined), had a significantly lower WTP than the two 'satisfied' groups combined ($U = 961.500$, $z = -1.960$, $p \leq 0.050$, $r = 0.16$). When visitors were asked if they would return to the Folkestone Marine Reserve in the future, the majority (80%) said they would, while the remainder said no.

Seawater (in terms of clarity) was rated highly by visitors, with a mean value of 4.48 ($\pm 0.43$ s.d.) recorded. Fish life was rated above average with a mean of 3.80 ($\pm 0.88$ s.d.).

**Table 6  Coefficient estimates of visitors' WTP using ordinary least squares (OLS) and interval (MLE) regression models.**

| Variable data | All data | All data | Artificial reef data | Artificial reef data | Natural reef data | Natural reef |
|---|---|---|---|---|---|---|
| Model | OLS | Interval (MLE) | OLS | Interval (MLE) | OLS | Interval (MLE) |
| Intercept | −6.542** | −5.958** | −7.719** | −7.30** | −9.401** | −8.958** |
| Age | −0.106*** (0.040) | −0.103*** (0.038) | – | – | −0.175*** (0.059) | −0.169*** (0.056) |
| Env_concern | 1.264*** (0.331) | 1.190*** (0.313) | 1.051** (0.428) | 1.00** (0.405) | 1.456*** (0.472) | 1.423*** (0.445) |
| Dived_FMR | −3.238* (1.771) | −3.149* (1.677) | – | – | – | – |
| Coral_life | – | – | – | – | 4.368*** (1.460) | 4.286*** (1.378) |
| Species_view | 5.806*** (0.516) | 5.685*** (0.490) | 5.052*** (0.709) | 4.99*** (0.672) | 6.573*** (0.714) | 6.422*** (0.677) |
| Model parameters | $n = 243$ F stat: 71.43 $p < 0.001$ $R^2$: 47%- | $n = 243$ Chi$^2$: 167.99 $p < 0.001$ | $n = 121$ F stat: 37.56 $p < 0.001$ $R^2$: 39% | $n = 121$ Chi$^2$: 61.6 $p < 0.001$ – | $n = 122$ F stat: 43.04 $p < 0.001$ $R^2$: 59% | $n = 122$ Chi$^2$: 112.21 $p < 0.001$ – |

**Notes.**

Standard errors in parentheses. Only significant variables shown.

*** Significance at the $p \leq 0.01$.

** Significance at the $p \leq 0.05$

* Significance at the $p \leq 0.10$

Coral life however, received the lowest mean rating of 3.26 (±0.99 s.d.). It was found that snorkellers and divers differed in their ranking of coral life, with snorkellers rating this attribute significantly higher than divers ($U = 5,510$, $z = −2.196$, $p \leq 0.028$, $r = 0.14$).

## Econometric analysis

The results of the ordinary least squares and interval regression models are presented in Table 6. Our results showed consistency in the coefficient estimations obtained between the two regression models, suggesting the payment card design used for the surveys was well ordered (*Cameron & Huppert*, *1989*) and/or the normality assumption was well maintained by the data (*Yang, Hu & Liu*, *2012*).

The explanatory powers of the ordinary least squares models were good, yielding $r^2$ values of 39%, or above (Table 6). Overall, five of the twelve estimated coefficients expected to influence WTP, were statistically significant. Based on previous research (*Arin & Kramer*, *2002*; *Lindsey & Holmes*, *2002*; *Seenprachawong*, *2003*; *Togridou, Hovardas & Pantis*, *2006*; *Hargreaves-Allen*, *2010*), variables expected to show significant explanatory power, but in the event did not, included number of years in education, previous catamaran trips and number of prior visits to Barbados. Of the variables found to be significant, three (*Age, Env_concern* and *Species_view*) were significant at the 1% level (*Env_concern* 5% significance level for the artificial reef survey), whilst *Dive_FMR* was marginally significant at the 10% level. Two variables (*Age* and *Dived_FMR*) had negative signs on the coefficients, implying that younger respondents and those who had not previously dived in the reserve were prepared to pay more as a daily Folkestone Marine Reserve fee. The coefficients for the

Kirkbride-Smith et al. (2016), *PeerJ*, DOI 10.7717/peerj.2175 — 15/32

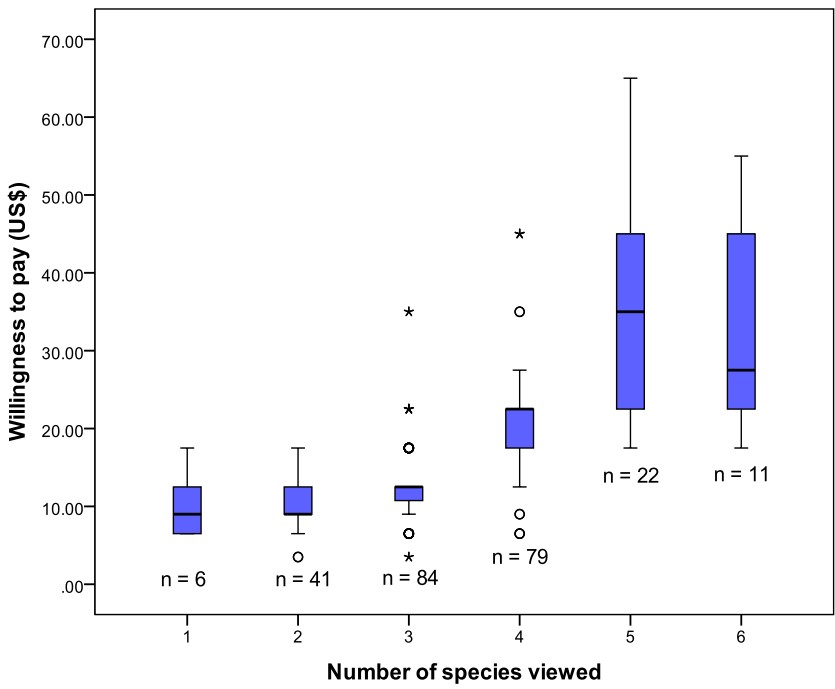

**Figure 3** The relationship between the number of marine species viewed and respondents WTP for reef protection in the Folkestone Marine Reserve (the line is the median, boxes the 25–75% quartiles and the whiskers the 95% CI).

remaining three variables (*Env_concern*, *Coral_life* and *Species_view*) were positive. This indicates that respondents who rated the coral life as good, reported higher levels of concern for the reefs and marine environment and viewed more marine life, had higher WTP. It should be noted, the variable *Coral_life* was only significant in the natural reef model.

The regression results indicated the variable '*Species_view*' made the largest unique contribution to the variance in WTP, with a mean value of 22% noted across all data sets. A one unit increase elevates WTP on average US$5.69–US$5.81 for each additional species viewed (Table 6).

A Kruskal–Wallis Test indicated a high level of association between the dependent variable and *Species_view* ($x^2$ (5) $= 133.39$, $p \leq 0.001$) (Fig. 3). Further post hoc analysis confirmed significant differences in WTP occurring between 'two and three' species viewed, 'three and four' species viewed and 'four and five' species viewed ($U = 1,119$, $z = -3.391$, $p \leq 0.001$, $r = 0.30$; $U = 1,154$, $z = -7.380$, $p \leq 0.001$, $r = 0.58$; $U = 314$, $z = -4.703$, $p \leq 0.001$, $r = 0.47$), respectively.

## DISCUSSION

The principal focus of this study was to estimate visitors' consumer surplus for a MPA in Barbados and to differentiate between visitors use values of natural and artificial reefs. As far as we are aware, it constitutes the first work to compare use values of two types of reef habitat within a reserve environment.

It is apparent from our results, that marginal (not significant) differences occurred between visitors WTP for natural reefs (US$18.33) and estimates for artificial reef use (US$17.58). Three studies (*Johns et al.*, *2001*; *Johns*, *2004*; *Oh, Ditton & Stoll*, *2008*) have reported use values relating to consumer's surplus of both reef habitats, and all three investigations yielded higher estimates for natural reef usage. *Oh, Ditton & Stoll* (*2008*) estimated an average consumer surplus for diving per trip in Texas waters at US$171 for natural reef divers and US$101 for artificial reef divers; a net increase of 70% per trip for scuba diving at natural reefs. Both *Johns et al.* (*2001*) and *Johns* (*2004*) estimated consumer's surplus for managing and maintaining the natural and artificial reefs in southeast Florida and Martin County, Florida, respectively. *Johns et al.* (*2001*) reported an average use value for residents and visitors at natural reefs of US$12.74/person-day and US$8.63/person-day for artificial reefs, at the same location. In a later study, (*Johns*, *2004*) estimated non-local tourists use value for diving, fishing and snorkelling combined at US$46.00/person-day at natural reefs, compared to US$23.84/person-day at artificial reefs.

It is suggested that a hypothetical bias linked to the 'warm glow' effect (*Andreoni*, *1990*; *Christie*, *2007*) may partially account for similar bid values been elicited for both reef types that we investigated. Other environmental studies have identified this phenomenon of impure altruism (*Nunes & Schokkaert*, *2003*; *Polak & Shashar*, *2013*), which may be more prevalent among tourists on vacation (*Polak & Shashar*, *2013*). Indeed, *Kahneman & Knetsch* (*1992*) propose that contingent valuation responses reflect WTP for the moral satisfaction of contributing to public goods—not the economic value of the goods in question, though most (75%) visitors in this present survey exhibited the motivation of bequest value as the main driver of WTP. In reality, (*Diamond & Hausman*, *1994*) believe that WTP would be more conservative if one were asked to pay for it during the surveying process. In spite of this, given at the time the Tiami cruise cost US$85 per person, it may be plausible that some respondents may have rounded their WTP up to US$100 regardless of the reef habitat being valued. In fact, 45% of bid values fell within the US$10–20 intervals (Table 2).

Several variables were significant in influencing WTP. We found that as respondent's age decreased bid value increased, which is not unusual in this type of study. *Arin & Kramer* (*2002*) also noted that younger people were more willing to donate towards reef conservation and *Uyarra, Gill & Côté* (*2010*) found that younger divers had a more positive attitude towards paying higher marine park entrance fees in Bonaire. Moreover, *Asafu-Adjaye & Tapsuwan* (*2008*) reported that Thai respondents accepted the bid in a contingent valuation study more readily as the age of the diver decreased. With regard to older generations, it may be plausible that they are more skeptical about contributing towards conservation efforts in general, or perhaps are more familiar and experienced with the goods being valued, thus reflecting reduced utility and diminishing marginal returns. In fact, we found repeat visitors to the reserve, had a significantly lower bid value than first-time visitors there. This result lends support to *Dharmaratne, Sang & Walling* (*2000*) who noted repeat visitors to a terrestrial park and marine reserve in Barbados and Jamaica respectively, had a lower WTP than first-time visitors. The present study also confirmed that environmental awareness and concern for reefs generally, had a positive

effect on payment bids, a trend confirmed in other WTP reef studies (*Tapsuwan*, *2006*; *Togridou, Hovardas & Pantis*, *2006*; *Casey, Brown & Schuhmann*, *2010*; *Hargreaves-Allen*, *2010*), though not consistent with *Barker*'s (*2003*) results.

Overall, the number of species viewed had the strongest effect on mean bid value for the marine park fee. The model indicated that each additional species viewed elevated WTP by approximately US$5.70 (Table 6). This suggests visitors are prepared to pay a significant amount to view wildlife within Folkestone. Indeed, marine life is regarded as one of the greatest sources of revenue for the dive and snorkel tourism industries (*Barker*, *2003*) and viewing it has a positive impact on customer satisfaction (e.g., *Musa*, *2002*; *Musa, Kadir & Lee*, *2006*; *Coghlan*, *2012*). WTP studies have shown that divers will pay significantly for conservation efforts that favour high biodiversity on artificial coral reefs (*Polak & Shashar*, *2013*) and for greater fish abundance/size on natural reefs (*Rudd & Tupper*, *2002*; *Barker*, *2003*; *Wielgus et al.*, *2010*). Individuals also hold considerable consumer surplus value for viewing large species such as dolphins, rays, whale sharks and turtles (*Davis & Tisdell*, *1999*; *Schuhmann, Casey & Oxenford*, *2008*; *Hargreaves-Allen*, *2010*; *Schuhmann et al.*, *2013*; *Farr, Stoeckl & Beg*, *2014*). In Barbados, turtles provide an additional means to attract tourists to the island (*Troëng & Drews*, *2004*; *Uyarra et al.* (*2005*), being widely promoted in various advertising campaigns. WTP to view turtles is substantial in this area of the Caribbean. *Schuhmann et al.* (*2013*) found divers in Barbados are prepared to pay over US$57 for the first encounter with a marine turtle, and approximately US$20 per 2-tank dive for each additional encounter. We also established that turtles are a valuable resource, as they were associated with an US$8 increase in mean bid value per person, compared to respondents who had not viewed a turtle during their trip.

Another important aspect of this research was to solicit visitors' opinions on reef material preferences for future purpose-built reef (Supplemental Information). Overwhelmingly, underwater art as sculptures was viewed as the most unappealing material choice. This is despite its reported success in marine parks in Cancun, Mexico and Grenada in the Caribbean (www.underwatersculpture.com). Salient points noted as to visitors general dislike of this type of reef appeared to firmly centre on the lack of available habitat for species refuge, such as holes and crevices for fishes, and also on the 'out of context' appearance of human statues underwater, as well as the small ecological footprint created. On the other hand, Reef Balls™ (www.reefball.org) presented as a snorkel trail, were viewed more favourably, especially among non-divers. Interestingly, (*Ramos et al.*, *2006*) concluded that concrete modules were the least important choice of reef material among scuba divers in Portugal. Nevertheless, snorkel trails have been used with notable success in parts of the Caribbean. For example, in Antigua a 5-row Reef Ball™ breakwater structure (Supplemental Information) also acts as a successful nature trail for snorkellers and divers (*Kaufman*, *2006*), and in the US Virgin Islands, nearly 90% of the 50,000 annual visitors use a managed snorkel trail (*Thorsell & Wells*, *1990*). Of significance, (*Hannak et al.*, *2011*) established that most visitors to a snorkel trail in Dahab, Egypt were willing to pay US$14–27 for a guided trip. Notwithstanding, purposefully sunken ships were found to be the most popular material choice among 73% of respondents. In previous studies (*Ditton et al.*, *2002*; *Stolk, Markwell & Jenkins*, *2005*; *Shani, Polak & Shashar*, *2011*; *Kirkbride-Smith, Wheeler*

& Johnson, 2013), divers have communicated an immense preference for shipwrecks and deliberately sunken vessels for artificial reef creation Content analysis of our data suggests the appeal of sunken ships is related to their perceived capacity to provide adequate substrate and shelter for marine species, their 'in keeping' generic form and visual appeal, and to their historical fascination.

## Policy recommendations

Our results demonstrate that almost all (97%) visitors would be willing to pay a Folkestone Marine Reserve user fee. By combining data of the artificial and natural reef models, our results indicate overseas tourists would be prepared to pay an average of US$18 as a fee per visit, which could supplement reef conservation finance. This amount is broadly consistent with the results of similar WTP studies (*Barker*, *2003*; *Mathieu, Langford & Kenyon*, *2003*; *Tapsuwan*, *2006*; *Hargreaves-Allen*, *2010*) and is in fact, well above the hypothetical fee structure proposed by the *Inter-American Biodiversity Information Network* (*2010*) for the marine reserve in Barbados. In this latter report, a fee of US$3 to US$5 for a snorkel tag, and US$5 to US$10 for a diving tag, is suggested. Data from this present study could therefore be used to aid the setting of a single, daily, user fee for Folkestone.

Implementing a successful fee system needs cooperation among visitors, tour operators and managers (*Terk & Knowlton*, *2010*). To encourage visitors' adoption of fees, they require clarity on how their money is used and managed (*Peters & Hawkins*, *2009*). Studies suggest that user fee acceptance improves if visitors have knowledge that their funds are managed appropriately (*Casey, Brown & Schuhmann*, *2010*) and specifically; that money is spent on reef protection (*Casey, Brown & Schuhmann*, *2010*) and on improving park management (*Yeo*, *2005*). In this study, we found participants concerned over how funds would be used and managed and established that three quarters of visitors wanted a non-governmental organization to manage their payments. To gain support in a fee system, supplying park booklets to visitors detailing the purpose and nature of fees, may assist. In fact, many respondents requested information about the biological aspects of the reserve, as did divers and snorkellers studied by *Barker* (*2003*) in St. Lucia. Moreover, by providing meaningful information for tourists, it helps develop place attachment and stewardship (*Ham*, *1992*). Snorkel and dive tour operators also need encouragement to adopt fees. As an incentive to collect them, *Terk & Knowlton* (*2010*) suggest a system for compensating operators administration time, by giving them a small percentage of the fees gathered. This system was originally employed in Mexico and appears a simple but fair approach.

Visitors also need to see 'what they are getting for their money' and good reserve infrastructure helps justify fee payment (*Sedley Associates Inc., AXYS Environmental Consulting (Barbados) Inc. & Scantlebury and Associates Ltd.*, *2000*). This is especially relevant to repeat customers who were noted as having lower WTP (Tables 5 and 6). Developing eco-tourism opportunities via artificial reefs can create unique selling points in a resort (*Dowling & Nichol*, *2001*; *Leeworthy, Maher & Stone*, *2006*; *Shani, Polak & Shashar*, *2011*; *Edney*, *2012*) and have the potential of drawing visitors to reserves. In previous research (*Kirkbride-Smith, Wheeler & Johnson*, *2013*) we established that artificial reefs were a prime motivator for some dive tourists to holiday in Barbados. Also, as fish

abundance is often greater within protected waters (e.g., *Chapman & Kramer*, *1999*; *Caselle et al.*, *2015*; *Sciberras et al.*, *2015*) it appears a fitting environment to deploy artificial reefs for amenity enhancement. Creating a new reef within Folkestone's waters appeared to be very popular among respondents, as over three quarters of those interviewed said this type of resource would encourage repeat visitation. We also discovered that many visitors had heard of artificial reefs and over a third had either snorkelled or dived on one previously, including many deployed in Barbados. Increasingly, artificial reefs are becoming more popular, especially among scuba divers (e.g., *Blout*, *1981*; *Scuba Travel*, *2006*; *Edney*, *2012*; *Kirkbride-Smith, Wheeler & Johnson*, *2013*), and given the substantial use value we report for them, it suggests visitors would be willing to support a reef substitution policy in Folkestone and potentially in other reserves offering this type of amenity.

Among the recreationally used natural reefs within the Folkestone Marine Reserve, it is the fringing reefs that are the most impacted (*Bell & Tomascik*, *1993*; *Lewis*, *2002*; *Inter-American Biodiversity Information Network*, *2010*) and this would appear the most appropriate zone to site underwater attractions. Several benefits could be yielded from developing artificial reefs in reserves. For example, managers may use them to influence and contain visitor use. Creating 'honey pot' sites within marine parks has been endorsed by some managers (*Clark et al.*, *2005*) as a strategy to conserve other coral reefs by redirecting reef use. Such a policy would be especially useful for managing in-training and novice divers who are documented as causing substantial damage to natural reefs (*Roberts & Harriott*, *1994*; *Walters & Samways*, *2001*; *Warachananant et al.*, *2008*; *Chung, Au & Qui*, *2013*). Moreover, these installations could be of value to snorkel and dive companies to help sustain existing local resources. However, concentrating tourist use is open to debate as (*Barker*, *2003*) found that visitors disliked the idea of being 'contained,' suggesting it would lead to overcrowding and reduced naturalness of an area. In contrast, (*Hannak et al.*, *2011*) established that a marine viewing trail would be the principal reason that their study group would choose a snorkel or dive site.

Notwithstanding, artificial reefs have been shown to offer opportunities to view interesting marine life (*Wilhelmsson et al.*, *1998*; *Perkol-Finkel & Benayahu*, *2004*; *Arena, Jordan & Spieler*, *2007*; *Kirkbride-Smith, Wheeler & Johnson*, *2013*). Indeed, studies have confirmed that artificial reefs can support a comparable diversity and density of marine species than are found on natural reef outcrops (*Clark & Edwards*, *1999*; *Perkol-Finkel & Benayahu*, *2004*), and this is especially true for fish abundance, where in some instances it has exceeded that present on natural reefs (*Fast & Pagan*, *1974*; *Wilhelmsson et al.*, *1998*; *Arena, Jordan & Spieler*, *2007*; *Santos, Oliveira & Cúrdia*, *2013*; *Granneman & Steele*, *2014*). Clearly, creating the right type of artificial reef that encourages a diverse species community is crucial for reef tourism, as this study showed the principal driver of WTP was marine life. In addition, artificial reef development allows for increased accessibility of reefs (*Milton*, *1989*; *Stolk, Markwell & Jenkins*, *2005*) and arguably, encourages the employment of more robust/resistant environments within reserves (*Marion & Rogers*, *1994*; *Claudet & Pelletier*, *2004*). To this end; MPAs provide the greatest opportunity to manage tourism use of natural reefs (*Thurstan et al.*, *2012*) and environmental enhancement using 'well planned' artificial reef could potentially facilitate this (*Oh, Ditton & Stoll*, *2008*).

## CONCLUSIONS AND FURTHER RESEARCH

This study focused on a MPA in Barbados to differentiate between respondents use values of natural and artificial reefs. Our findings show that most visitors are prepared to pay for reef conservation in the Folkestone Marine Reserve, and this represents an unexploited revenue stream that could be used for its day to day management. A mean WTP of US$18.33 and US$17.58 was estimated for natural and artificial reef use, respectively. This latter information could aid the setting of a single, daily, user fee for the islands marine reserve. Of importance, our results additionally indicate that significant use value could be gained from the provision of recreation-orientated artificial reefs within a reserve environment like Folkestone.

This research serves as a valuable foundation for future work that should aim to uncover divers' WTP for 'diving trips' within Barbados's MPA. Also, cruise trip passengers were not represented in this current study, and ideally, this omission needs addressing in future WTP studies for Folkestone. Finally, research into the recovery of non-use values (not current users of the resource) to fund reef management in the Folkestone Marine Reserve, is also an area worthy of future exploration.

## ACKNOWLEDGEMENTS

Our sincere thanks and gratitude extend to Denis Roach, the owner and Chief Executive Officer of Tiami Catamaran Cruises, and to his excellent staff for supporting the data collection period. We thank Jamar Archer, Thomas Atwell, Noddy Banfield, Michael Brown, Michael Captain, Diego De Beauville, Peter Hoad, Ryan Hoyte, Shea Innis, Joshua Roach, Roger Scandella, Sam Sealy and David Young. Additionally, our thanks go to Jeffrey Smith who helped with the data collection and to the reviewers whose helpful comments have greatly improved the final paper. Finally, we thank Todd Barber, Will Brown and Jason deCaires-Taylor for their permission to use images featured in the Supplemental Information and all survey participants who generously gave their time.

### Funding

The authors received no funding for this work.

### Competing Interests

Magnus L. Johnson is an Academic Editor for PeerJ.

### Author Contributions

- Anne E. Kirkbride-Smith conceived and designed the experiments, performed the experiments, analyzed the data, contributed reagents/materials/analysis tools, wrote the paper, prepared figures and/or tables, reviewed drafts of the paper.
- Philip M. Wheeler analyzed the data, reviewed drafts of the paper.

- Magnus L. Johnson conceived and designed the experiments, analyzed the data, contributed reagents/materials/analysis tools, wrote the paper, reviewed drafts of the paper.

## Human Ethics

The following information was supplied relating to ethical approvals (i.e., approving body and any reference numbers):

All participants completed the survey themselves and gave their permission to use the results. Individuals were not identifiable from the data provided. The work described in this paper was reviewed and approved by the Centre for Environmental and Marine Science departmental ethics committee (certificate number H030). Verbal assurance was provided by a representative of the Barbadian Coastal Zone Management Unit that no permit is required to conduct questionnaire based research on the island.

## Data Availability

The raw data has been supplied as Supplemental Information.

## Supplemental Information

Supplemental information for this article can be found online at http://dx.doi.org/10.7717/peerj.2175#supplemental-information.

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
