# Peer review of "Artificial reefs and marine protected areas: a study in willingness to pay to access Folkestone Marine Reserve, Barbados, West Indies"

_PeerJ, doi:10.7717/peerj.2175_

## Round 0.1 · original submission · Minor Revisions

I am sorry for the considerable delay in getting this back to you - we had difficulty getting referees to respond over the holiday season, and just got the second review returned on your manuscript now. Both referees are supportive of your work with a few relatively minor comments for clarity, content or style for you to consider in your revisions. I agree with the majority of suggested changes, but they all appear relatively straightforward to me, so I do not expect you should have much difficulty with addressing them, and I look forward to seeing your revised manuscript.

·

Basic reporting

This manuscript passes all the areas, except the figure quality, on which I have some substantial comments.

Experimental design

Very good experimental design, addresses the questions they set out to answer; it is relevant and meaningful. There are high technical standards applied to the experimental design and the data analysis. The experiment appears to have been conducted within ethical standards in research on human subjects.

Validity of the findings

Data appear to be statistically sounds, robust, and controlled.

Additional comments

This is a valuable study and raises some points which will encourage more research into this area of willingness to pay studies, the findings from which can effectively then be used to design dive fees that can be helpful for reef management. Novel study for quantifying and comparing both willingness to pay for natural reef viewing as well as for artificial reefs. Well-designed and statistically sound work. Insightful discussion, showing clear knowledge of the relevant literature and also how this study fits in with the pre-existing literature and moves our collective knowledge forward. I have some comments, which are mainly stylistic and a few content comments; they can be found in comment bubbles in the attached PDF file. But those should be easy to address. Very good work overall!

Reviewer 2 ·

Basic reporting

There seems to be some confusion as to whether this is a donation or a user fee. Is it a donation to the park to manage and protect corals or is it a user fee to go snorkeling. I think it is important to clarify and to be consistent throughout the paper. Also, the user of the word "diver" and the information provided about dive quality and dive experience make it confusing as the analysis is only for snorkel trips. I also think the information in the introduction needs to be updated as the sources are, in some cases, up to 16 years old.

Experimental design

My only concern is with the econometric analysis. Given the interdisciplinary nature of the audience it seems to me that you need more theory to inform the empirical model and the variables chosen. I would like to see the full model based on solid economic theory and at least report those results in a footnote in addition to the models reported. I would also like to see the difference between SCUBA and non-SCUBA participants and their Max WTP. The overall satisfaction with the trip and turtles seem like important variables of interest and should be included in the regression analysis.

Validity of the findings

I think that by emphasizing the potential problems and biases you provide a reason to be very skeptical of your results and possibly a reason to reject the paper. Perhaps you could look at the distribution of responses and tell us more about the reasons people gave and emphasize that less than half the bids fell between 10 and 20 dollars.

You may also want to have two results sections - one for the WTP analysis and another for the reef material analysis. Both pieces are important but for expository reasons it may be helpful to have a distinct separation. Also, I would recommend having a 4.1 "Policy Recommendations" section. In that section I do not think you should suggest a $10 fee - let the analysis speak for itself.

Additional comments

I like this paper very much and think it will add to our understanding of user fees as a potential source of revenue for marine conservation in the Caribbean. With a little work, I think it should be published. I am attaching a copy of the paper with my hand-written notes in hopes this will make it easier for you to make the changes I have suggested. I would like to see the revised version of the paper, but if the editors feel comfortable with the changes I am willing to forgo reviewing the paper again.

Annotated reviews are not available for download in order to protect the identity of reviewers who chose to remain anonymous.

---

## Round 0.2 · accepted · Accept

Having read through your resubmission, I see that you have incorporated nearly all the suggested changes. Both referees were positive about your initial submission, and you have addressed the comments of each to my satisfaction with this revision. Therefore, I am happy to accept your manuscript and move it forward into production at this time.